# Non-Canonical Programmed Cell Death in Colon Cancer

**DOI:** 10.3390/cancers14143309

**Published:** 2022-07-07

**Authors:** Bingchen Pan, Bowen Zheng, Chengzhong Xing, Jingwei Liu

**Affiliations:** Department of Anus and Intestine Surgery, First Affiliated Hospital of China Medical University, Shenyang 110001, China; bcpan@cmu.edu.cn (B.P.); 20201023@cmu.edu.cn (B.Z.)

**Keywords:** cell death, carcinogenesis, colon cancer

## Abstract

**Simple Summary:**

Non-canonical PCD is an important player in colon cancer cell suicide. It influences colon cancer in many ways, such as through tumorigenesis, treatment, and prognosis. In this review, we present the mechanism, application, and prospect of different types of non-canonical PCD in colon cancer.

**Abstract:**

Programmed cell death (PCD) is an evolutionarily conserved process of cell suicide that is regulated by various genes and the interaction of multiple signal pathways. Non-canonical programmed cell death (PCD) represents different signaling excluding apoptosis. Colon cancer is the third most incident and the fourth most mortal worldwide. Multiple factors such as alcohol, obesity, and genetic and epigenetic alternations contribute to the carcinogenesis of colon cancer. In recent years, emerging evidence has suggested that diverse types of non-canonical programmed cell death are involved in the initiation and development of colon cancer, including mitotic catastrophe, ferroptosis, pyroptosis, necroptosis, parthanatos, oxeiptosis, NETosis, PANoptosis, and entosis. In this review, we summarized the association of different types of non-canonical PCD with tumorigenesis, progression, prevention, treatments, and prognosis of colon cancer. In addition, the prospect of drug-resistant colon cancer therapy related to non-canonical PCD, and the interaction between different types of non-canonical PCD, was systemically reviewed.

## 1. Introduction

Programmed cell death (PCD) is an evolutionarily conserved process of cell suicide, which is regulated by various genes and the interaction of multiple signal pathways [1]. Programmed cell death (PCD) is well known as an essential regulator of various biological processes, especially in the carcinogenesis and therapy of tumors. There are multiple types of PCD, including apoptosis, mitotic catastrophe, parthanatos, ferroptosis, pyroptosis, necroptosis, oxeiptosis, NETosis, PANoptosis, entosis, autophagy, lysosome-dependent cell death, and alkaliptosis. Apoptosis is a well-known canonical PCD, so the remaining types constitute non-canonical PCD. Yet, we excluded apoptosis in this review because apoptosis is resisted by cancer cells almost systematically. Thus, non-canonical PCD is considered as a more prospective field [2] and a more fascinating alternative target [3]. In addition, there are overmuch studies about autophagy and inadequate studies about lysosome-dependent cell death and alkaliptosis. Therefore, we mainly discuss the remaining nine types of non-canonical PCD.

Colon cancer is the third most incident and the fourth most mortal despite significant progress that has been made for the treatment of colon cancer, the prognosis of colon caner still remains unfavorable [4]. The five-year survival rate for early-stage colon cancer patients is more than 90%, yet only approximately 10% for the later stages [5]. Multiple factors such as alcohol, obesity, and genetic and epigenetic alternations contribute to the carcinogenesis of colon cancer [6]. An increasing number of studies suggest that non-canonical PCD plays an essential role in the tumorigenesis, progression, prevention, treatment, and prognosis of colon cancer. Moreover, non-canonical PCD participates in tumor-leading inflammatory bowel diseases and anti-tumor effects in cancer therapy [7]. Here, we reviewed recent advances concerning the regulation of non-canonical PCD in multiple aspects of colon cancer.

## 2. Mitotic Catastrophe and Colon Cancer

### 2.1. The Mechanism of Mitotic Catastrophe

Mitotic catastrophe (MC) is a type of programmed cell death. It is known as a procedure leading to premature mitosis by the disfunction of the DNA structure checkpoint and spindle assembly checkpoint (Figure 1). MC usually results in with apoptosis, necrosis, or senescence. Mitotic catastrophe is mediated by genes such as the proto-oncogene RBM3 [8], the tumor suppressor genes p53 [9] and p21 [10], and proteins such as RNA binding protein Musashi-1 [11] and CUGBP2 [12]. Radiation therapy and drugs that damage microtubules can induce MC [13]. The inhibition of mitotic catastrophe might contribute to oncogenesis [14].

Single (SSBs) or double (DSBs) strand breaks often lead to mitotic catastrophe. ATM and ATR can activate checkpoints and target Chk1 and Chk2. Chk1 and Chk2 down regulate cdk1/CyclinB, cdk2/CyclinE, and cdk2/CyclinA and cause mitotic catastrophe. Three types of aberrant cell division can lead to mitotic catastrophe, and they can switch from one type to another [15].

### 2.2. The Relationship of Mitotic Catastrophe with Initiation and Progression of Colon Cancer

The well-known tumor suppressor gene CELF2 plays an essential role in colon cancer initiation and progression. The expression of CELF2 keeps decreasing during neoplastic transformation. The ectopic overexpression of CELF2 suppresses tumors by inducing cell death due to mitotic catastrophe [16]. The loss of KLF14, a tumor suppressor gene, is also associated with tumorigenesis [17]. As a member of the inhibitor of the apoptosis (IAP) gene family, survivin is expressed in a wide range of common types of cancer. The survivin mutant inhibits tumorigenesis and progression in colon cancer [18]. Proto-oncogenes such as RBM3 prevent mitotic catastrophe; thus, they promote the initiation and progression of colon cancer [8]. Interestingly, mitotic catastrophe plays a dual role in colon cancer initiation. During mitotic catastrophe, aneuploid cells may appear and cell DNA may be damaged, leading to oncogenic circumstances. Thus, incomplete mitotic catastrophe increases tumorigenesis [18]. Yet, more studies about this mechanism in colon cancer are required. Inflammatory bowel diseases (IBD) increase the risk of colorectal cancer [19]. Increasing mitotic catastrophe in normal colon cells leads to IBD [20]. Thus, mitotic catastrophe can also promote tumorigenesis of colon cancer.

### 2.3. Anti-Tumor Treatments Inducing Mitotic Catastrophe

Nuclear lamina constitutes the skeleton of cell nuclear and influences tumor adhesion and migration. It has been discovered that upregulated LMNB1 can induce MC in colon cancer cells after 5-FU treatment. Thus, upregulating LMNB1 might be helpful in colon cancer therapy [21]. A lot of other drugs can kill colon cancer cells via mitotic catastrophe, such as thieno pyrimidine derivatives 6j [22], a low dose of flubendazole [23], stilbene 5c [24], imidazoacridone C-1311 [25], and quinoline derivative [26].

Chemoresistance restricts the efficiency of chemotherapy, and mitotic catastrophe contributes to chemoresistance in some cases. When tumor cells are exposed to cisplatin in vitro, mitotic activity initially stops, but the cells maintain DNA replication. This kind of DNA replication produces huge polyploid cells, which then initiate aborted mitosis and may die from mitotic catastrophe. However, many polyploid cells survive and develop into extensive colonies. A recent study discovered that these colonies were made of small diploid cells, indicating de-polyploidy from polyploid cells to small diploid cells. A multi-step approach, including DNA internal replication, polyploidy, then de-polyploidy, and the generation of clone escape cells, can explain tumor recurrence after the initial effective chemotherapy [27]. The catalytic inhibitors of topoisomerase IIα cause colon cancer cells to undergo mitotic catastrophe, and the cancer cells show higher sensitivity to ICRF 193 [28]. It is observed that the PSAT1 expression level is negatively correlated with the drug response caused by the decrease of mitotic catastrophe [29].

The SAC (spindle assembly checkpoint) makes sure chromosomes segregate correctly to avoid MC, and the DNA structure checkpoint prevents DNA damage. There are drugs aiming at the SAC or DNA structure checkpoint to induce MC in colon cancer cells, including geraniin [30], docetaxel, and vincristine, which have a synergistic effect with Aurora inhibitor SNS-314 [31], a rimonabant that suppresses the progression of colon cancer [32] and low doses (~0.5 μM) of reversine [33].

Cell-cycle checkpoints cope with various DNA impairments. Chk1 is the vital mediator of cell-cycle checkpoints. The inhibition of Chk1 leads to mitotic catastrophe. UCN-01 induces the mitotic catastrophe of colon cancer cells by targeting Chk1 [34]. Some studies revealed that Chk1 status might influence the effectiveness of treatment. For example, cyclin B1 is a predictor of the effect of Chk1 inhibitors in numerous types of cancer, including colon cancer, which might be applied to the stratification of colon cancer [35]. Chk1 inhibition also enhances the effect of other drugs. It is discovered that Chk1 knockdown promotes mitotic catastrophe in colon cancer cells treated with paclitaxel [36]. Chk2 is the down-regulator of Cdk1/cyclin B. Thus, inactive Chk2 promotes mitotic catastrophe and then raises chemo-sensitivity [37].

Mitotic spindles and centrosomes play a vital role in mitosis. Glycogen synthase kinase (GSK) 3β locates in mitotic spindles and centrosomes. It can interact with TPR and promote the generation of the mitotic spindle. Therefore, the GSK3β inhibitor can cause mitotic catastrophe in colon cancer cells [38]. The overexpression of KLF14 leads to centrosome overduplication and then mitotic catastrophe [17]. It has been reported that a water-soluble phenanthridine targets supernumerary centrosomes to induce mitotic catastrophe and becomes a target therapy [39]. Drugs such as paclitaxel, 4-Hydroxy-2-Nonenal, and Aurora-A targeted drugs can also disturb the mitotic spindle checkpoint and lead to mitotic catastrophe [40]. As a treatment without drugs, there was no centrosome staining discovered after heat shock and mitotic catastrophe was induced [41]. Interestingly, mitotic catastrophe induced by thermal radiosensitizers can enhance the sensitivity of colon cancer cells to heat shock in turn [42].

The p53 gene inhibits tumor generation and progression, so multiple treatments depend on the status of p53. Some drugs have a better function under p53 + status. For example, low doses (~0.5 μM) of reversine kill wild-type colon cancer cells more efficiently than their p53- counterparts [33]. HCT116/PKCδ cells show symbols of mitotic catastrophe when p53 is upregulated [43]. Yet, some drugs can induce mitotic catastrophe even when the status of p53 is deficient. When DNA synthesis is interrupted, p53-deficient CHK1 tumor cells will induce premature mitosis, causing mitosis catastrophe [44]. PTX (paclitaxel), SP600125, and a low dose of DCQ can trigger mitotic catastrophe under p53-status [45]. Yet, sometimes mitotic catastrophe is independent of p53 such as with the case of mithramycin SK [46].

The deletion of survivin, a member of the inhibitor of apoptosis (IAP) family, is highly expressed in colon cancer tumor cells and leads to mitotic catastrophe [47]. Survivin is related to drug resistance, angiogenesis, and poor prognosis. The survivin mutant Thr34Ala [48] and Cys84Ala [18] suppress cell proliferation, induce mitotic catastrophe, and raise the sensibility to chemotherapy in oxaliplatin [48] and 5-FU [18]. Targeting survivin can be a form of gene therapy [18]. It is efficient to combine it with chemotherapeutics [48]. Interestingly, there is an interaction between survivin and chemotherapeutics. Oxaliplatin can inhibit survivin and induce mitotic catastrophe in return and can be a combination with other drugs [49].

Radiation can induce mitotic catastrophe in colon cancer cells, while mitotic catastrophe can sometimes enhance the effect of radiation therapy. As a radiosensitizer, the extract of Derris scandens Benth regulates radiosensitization through mitotic catastrophe [50]. Another radiosensitizer XL-844 exerts a similar effect [51]. In addition, the inhibition of mitotic catastrophe can lead to the resistance of colon cancer to radiation via the down-regulation of CUGBP2 [52].

### 2.4. Prevention of Colon Cancer by Mitotic Catastrophe

P53 is also related to the prevention of mitotic catastrophe. Heat-stable enterotoxin (ST) resists RIGS (acute radiation-induced GI syndrome) and prevents colon cancer. This procedure requires p53 activation to restrict mitotic catastrophe. Yet, the ability of p53 is limited. The GUCY2C–cGMP axis can prolong the cell cycle by delaying the G1-S phase by partly meditating p21. It prevents RIGS by restricting mitotic catastrophe and maintaining the integrity of the intestinal epithelial barrier. Meanwhile, colon cancer cells treated with cGMP show no significant difference in radiosensitivity compared to other colon cancer cells. Thus, they do not alter the radiosensitivity of colon cancer, which makes it a utility approach for preventing RIGS in radiotherapy and colon cancer initiation [53]. Instead of genes, an abundant amount of literature indicates the existence of an immunosurveillance system can control colon carcinogenesis. Tetraploid Tp53−/− colonocytes fail to survive in immunocompetent mice and develop neoplastic lesions in immunocompromised settings only. This phenomenon confirms the function of immunosurveillance [54]. Rimonabant can reduce the formation of precancerous lesions in the mouse colon and prevent tumorigenesis. Yet more studies about rimonabant in human cells are required [32]. In early tumorigenesis, mitotic catastrophe is resisted by the deacetylation-phosphorylation regulation of the SIRT2-SMC1A axis. The inhibition of SIRT2 activity or continuously increasing SMC1A-K579 acetylation causes abnormal chromosome segregation and induces mitotic catastrophe in cancer cells. The dysregulation of this procedure encourages early precursor lesions to overcome oncogenic stress [55].

## 3. Ferroptosis and Colon Cancer

### 3.1. The Mechanism of Ferroptosis

Ferroptosis is an iron-dependent mode of non-apoptotic cell death characterized by the iron-dependent [56] accumulation of lipid reactive oxygen species (ROS) [57] and lipid peroxidation [58] (Figure 2). The enzyme glutathione peroxidase 4 (GPX4) is the key regulator of ferroptosis [59]. Recent research discovered that enriched KEGG pathways induce ferroptosis through deregulating miRNA [60]. Ferroptosis plays a vital role in the treatment of cancer and can be an essential element in the prognosis model.

In the first signal pathway, Cys2 is transferred into the cell and conversed into Cys. Cys is a key component in GPX4 formation. CARS1 inhibits this procedure and therefore inhibits ferroptosis. In the second signal pathway, ACSL4 (cysteinyl-tRNA synthetase) plays a key role in PUFA-CoA composition and ferroptosis inducement. In the third signal pathway, Fe^3+^ is transferred into the cell through TFRC (transferrin receptor protein 1) and turns into Fe^2+^. Fe^2+^ constitutes LIP (labile iron pool) and increases the accumulation of ROS. Iron-responsive element-binding protein 2 (IREB2) also accumulates ROS. NCOA4 may not be a ferroptosis element. Instead, it may induce ferritinophagy.

### 3.2. Ferroptosis-Related Genes and RNAs in Colon Cancer

P53 is a well-known tumor suppressor, and it participates in plenty of antitumor procedures. N-acylsphingosine amidohydrolase (ASAH2), a kind of neutral ceramidase, is highly expressed in tumor-infiltrating MDSCs (myeloid-derived suppressor cells) of colon cancer. MDSCs can suppress the T cell immune response towards colon cancer cells. The inhibition of ASAH2 balances p53 protein and increases ferroptosis in MDSCs and slows down colon cancer growth [61]. YAP1 is a downstream target of p53. Both of them are upregulated by cytoglobin (CYGB). YAP1 is the key promotor of ferroptosis; thus, CYGB becomes a novel tumor inhibitor [57].

Ferroptosis-related genes (FRGs) play a vital role in the construction of the prognosis model of colon cancer. 15 ferroptosis-related genes (FRG) were used for designing a prognosis model recently. It is discovered that highly expressed CISD1, GSS, and FDFT1 are associated with better prognosis, but PEBP1 is related to poor prognosis in stage II colon cancer patients [62]. In another research, a novel model for colon cancer including 7 FRGs was designed. Four of them (NOX4, CARS, WIPI1, and CDKN2A) are promoters of ferroptosis, while the other three genes (TFAP2c, SLC2A3, and DRD4) protect cells from ferroptosis [63]. Interestingly, the FRG prognosis model is also related to immune status such as CD8+T cells and STING. STING is a novel target, and it deserves further research [64].

The ferroptosis-related lncRNA (frlncRNA) pair can also predict the prognosis of colon cancer. A recently designed model including 25 frlncRNA pairs [65] and a model with 7 frlncRNA [66] were proved as promising prognosis models. Ferroptosis-related lncRNAs can predict prognosis as well as reflect other conditions of colon cancer, including hypoxia condition; immune-related factors; somatic variants; and signaling pathways such as MAPK, mTOR, and the glutathione metabolism pathway. All of these models are based on the TCGA database, GEO, or the public FerrDb database, requiring more research to test the clinical significance.

### 3.3. Anti-Tumor Treatments Inducing Ferroptosis in Colon Cancer

#### 3.3.1. Anti-Tumor Treatments Inducing Ferroptosis

Some drugs can function by raising ROS. Bromelain is known as a ROS-raising drug. In Kras- colon cancer cells, bromelain upregulates ACSL-4, which induces ferroptosis by accumulating ROS. Thus, it might become a therapy for Kras mutant colon cancer [67]. SCHL, a naturally occurring indoloquinazoline alkaloid, can induce ferroptosis through increasing ROS. It deserves further study as a novel drug [68]. In addition, the deubiquitinase OTUD1 enhances iron transport mediated by transferrin receptor protein 1 (TFRC) through deubiquitinating and stabilizing IREB2. This procedure leads to increased ROS production and iron death in colon cancer cells [69].

There are also drugs related to GPX4. RB (resibufogenin) can trigger ferroptosis in a GPX4 inactivation-dependent mode. Thereby, RB can inhibit the progression of CRC cells and tumorigenesis [70]. HNK (honokiol), a biphenolic compound, can raise the levels of ROS and Fe^2+^. Additionally, it also decreases the activity of GPX4. Thus, HNK can induce ferroptosis in colon cancer cells [2]. H2S (hydrogen sulfide) plays a vital role in colon tumor proliferation and progression. Interestingly, besides being an MRI contrast agent, fusiform iron oxide-hydroxide nanospindles (FeOOH NSs) can also induce ferroptosis by eliminating H2S [71].

As for drug resistance of colon cancer, it has been found that Lipocalin 2, a kind of siderophore-binding protein, can inhibit ferroptosis and lead to drug resistance against 5-FU. Lipocalin 2 functions through decreasing the iron level and increasing GPX4 [72]. XCT and H2S are chemoresistance initiators. Mutual regulation between xCT and H2S can be suppressed by AOAA and Erastin. This combined therapy leads to ferroptosis in chemoresistance colon cancer cells [73]. It is well known that many saponins are anti-tumor drugs. Ardisiacrispin B, one of the saponins, can function by inducing ferroptosis and participating in multidrug-resistance colon cancer cells [74].

Myeloid-derived suppressor cells (MDSCs) can suppress the T cell immune response towards colon cancer cells. ASAH2, a kind of neutral ceramidase, can unbalance p53 protein and prevent ferroptosis in MDSCs. NC06, as the inhibitor of ASAH2, therefore might become a promising immunotherapy [61].

Ferroptosis can combine with other types of programmed cell death and enhance the effect of treatment. Sorafenib (SRF), one of the ferroptotic agents, induces not only ferroptosis but also apoptosis in human colon cancer cells. SRF increases the TRAIL (TNF-related apoptosis-inducing ligand) receptor DR5 through the CHOP pathway [75].

#### 3.3.2. Ferroptosis Activators

Hypoxia is a feature of solid tumors. Transcription factor HIF-2α (hypoxia-inducible factor 2α) plays a significant role in hypoxic responses. Yet, it can be resisted by cancer cells rapidly. It was discovered recently that HIF-2α promotes oxidative cell death by increasing cellular iron and ROS in colon cancer cells under the meditation of ferroptosis activator dimethyl fumarate (DMF). As the major transcriptional regulator of cellular iron levels, HIF-2α enhances the vulnerability of the proteome to oxidative damage with the existence of DMF and promotes ferroptosis [76]. Activating the p53 tumor suppression pathway is also a common choice in cancer therapy. Triterpene saponin derivatives D13, a p53 activator, and can induce ferroptosis in colon cancer cells without normal organ toxicity [77].

#### 3.3.3. The Role of Ferroptosis in Tumor Vaccines

An interesting discovery has been made about the tumor vaccine recently. The tumor vaccine induces the immune response against a tumor. Thus, it is a promising anti-tumor therapy. Yet, the effect can be suppressed because of the TME (tumor microenvironment). Additionally, the production of the vaccines made of exosomes is pretty low. As a vastly producible vaccine, nanovesicles (eNVs-FAP) are more applicable in clinic than exosomes. Astonishingly, eNVs-FAP can reprogram immunorestraining TME in colon cancer cells and weaken drug resistance. It can stimulate the immune response to promote ferroptosis by producing IFN− γ and consuming FAP + CAFs. Additionally, PAF is overexpressed in 90% of tumor cells. Thus, eNVs-FAP might be applied to tumor vaccines [78].

### 3.4. Contradiction of NCOA4 Effect

There is a contradiction about NCOA4-mediated ferritinophagy. NCOA4 is used to be considered as a promoter of ferroptosis [79,80], yet in a recent study, it was stated that NCOA4 disruption is irrelevant to ferroptosis in colon cancer cells. This may have been due to the different cell lines they used. In former studies, mouse embryonic fibroblasts (MEFs) and human HT-1080 fibrosarcoma cells were applied, yet colon cancer cells were used in the latter study. The intestinal tissue can balance the iron level and confront the function of NCOA4, which is the disturbing iron level. So, NCOA4 might not be valid in colon cancer treatment [81].

## 4. Pyroptosis and Colon Cancer

### 4.1. The Mechanism of Pyroptosis

Pyroptosis is a kind of programmed cell death that causes cells to expand ceaselessly until the cell membrane bursts, leading to the leak of cell contents and activating a strong inflammatory response (Figure 3). It has recently been found that gasdermin D (GSDMD) and GSDME are incised [82] by active caspase-1/or caspase-4/5/11 [83], and caspase-3. The N-terminal of gasdermin protein is cut off [84] from the C-terminal repressor domain (RD). Additionally, the N-terminal leads to pyroptosis by piercing the cell membrane. The swelling cells and leakage of intracellular contents are mediated by this hole-forming procedure [85]. The leakage can release immunogenic-damage-related molecular patterns (DAMPs), including the well-known subtype HMGB1. HMGB1 becomes the marker of pyroptosis [86]. Therefore, pyroptosis is gasdermin-mediated pro-inflammatory cell death [87]. Recent evidence suggests that pyroptosis is an important regulator of colon cancer in many aspects, including prevention and treatment.

LPS (lipopolysaccharide) modifies caspase4/5/11 to become active. Gasdermin D (GSDMD) is incised by active caspase-1/or caspase-4/5/11. The N-terminal of gasdermin protein is cut off from the C-terminal repressor domain (RD). Additionally, the N-terminal leads to pyroptosis by piercing the cell membrane.

### 4.2. Pyroptosis-Related Genes in Colon Cancer

The NLRP (NOD-like receptor(pyrine)) gene is an essential gene in pyroptosis. The highly expressed NLRP gene can activate caspase-1 and lead to pyroptosis. Mice with the deficient NLRP gene and the caspase-1 gene are less likely to acquire colitis-related colon cancer [88]. Varies types of NLRP genes play roles in pyroptosis. The mutational ITH of *NLRP9* is common in the CRC, and it correlates with the poor prognosis of colon cancer. NLRP9 is altered at multiple levels (frameshift mutation, mutational ITH, and loss of expression), which together could contribute to the pathogenesis of CRC [89]. The GSDME gene is a vital gene in pyroptosis. Knocking out GSDME switched lobaplatin-induced cell death from pyroptosis to apoptosis [90]. GSDME(−/−) (also known as Dfna5(−/−)) mice were protected from chemotherapy-induced tissue damage and weight loss. Thus, the GSDME gene is highly related to chemo-resistance [83].

Pyroptosis-related genes can also predict the prognosis of colon cancer. In a recent study, 13 genes (KIF7, SYNGR3, NCKAP5L, ZKSCAN2, SIX2, OLFM2, GPSM1, ZEB1-AS1, CD72, TGFB2, CSRP2, TRPV4, and LHX6) constituted a prognosis model [91]. In another study, eight genes, including cytotoxic T-lymphocyte-associated protein 4 (CTLA4), chemokine (C-C motif) ligand 11 (CCL11), ninein (NIN), transmembrane protein 154 (TMEM154), kinesin family member 7 (KIF7), KIAA1671, ribonuclease P/MRP 14-kDa subunit (RPP14), and cadherin19 (CDH19), were identified as the key genes in the prognosis model [92].

### 4.3. Anti-Tumor Treatments Inducing Pyroptosis

The caspase-1/caspase-4/5/11-GSDMD pathway is one of the main pathways of pyroptosis. Treatments aiming at this pathway are numerous. Caspase-1 can be upregulated by protocatechuic acid (PA) at a higher dose (25 and 50 lg/mL). It indicates the potential of inducing pyroptosis in colon cancer by PA. Yet, more specific research is required to confirm this conjecture [93]. Conjugated linolenic acid (CLNA) is a kind of ω-3 fatty acid. CLNA1 and CLNA2, two conjugated isomers of CLNA, can lead to pyroptosis by activating caspase-1 and caspase-4, 5 and induce colon cancer cell pyroptosis [94]. Liver X receptors (LXRs) are possible anti-tumor targets. The location of LXRβ is correlated with the activation of caspase-1 and pyroptosis under the treatment with the LXR ligand [95]. Two more pieces of research confirm that LXRβ can induce pyroptosis by activating caspase-1 [96,97]. These discoveries enhance the assumption of the validity of LXRβ being a target in colon cancer therapy. NALP1, a member of the nucleotide-binding oligomerization domain-like receptor family, induces pyroptosis in colon cancer cells. NALP1 and caspase-1 are connected by ASC (apoptosis-associated speck-like protein containing a CARD) [98]. Secretoglobin (SCGB) 3A2 can activate caspase-11 and induce pyroptosis in colon cancer. All the sensitive cells express caspase-1 and caspase-4, which indicates the potential activation ability of SCGB3A2 [99]. A type of nanostructured toxin T22-DITOX-H6 increases caspase-11 expression in colon cancer cells, indicating that it induces pyroptosis [100].

Caspase-3-GSDME is another crucial pathway, and treatments aiming at it are vital. Oxaliplatin plus GW4064 can induce pyroptosis through the BAX/caspase-3/GSDME pathway and inhibit tumor growth in vivo. The synergistic effect between them solves the drug resistance of colon cancer against oxaliplatin [101]. Caspase-3 can also be activated by lobaplatin, leading to GSDME cleavage. It induces ROS/JNK signaling and then induces pyroptosis. This discovery is significant for the clinical application of colon cancer therapy [90].

In addition, there are treatments that induce pyroptosis through special pathways. The stimulator of interferon genes (STING) is a vital element in the immune response. It induces pyroptosis via spleen tyrosine kinase (Syk) in ex vivo experiments. The STING/Syk pathway serves as a novel colon cancer therapy strategy. It deserves more research in vivo [102].

### 4.4. Prevention of Colon Cancer by Inhibiting Pyroptosis

As a bowel disease characterized by inflammation, ulcerative colitis (UC) has been reported to correlate with the tumorigenesis of colon cancer [7]. The partial reversal of pyroptosis can be done by YST (Yu Shi An Chang Fang), which protects the intestinal mucosal barrier and prevents tumorigenesis [103]. GSDME-mediated pyroptosis releases HMGB1 and leads to colitis-associated colorectal cancer through the ERK1/2 pathway. Blocking GSDME is remarkably associated with a better prognosis in mice [87].

### 4.5. The Role of Caspase-2 in Pyroptosis of Colon Cancer

Despite multiple caspases being studied thoroughly during more than 20 years of research, the biological function of caspase-2 is still poorly known. Caspase-2 activity and caspase-2-induced MDM-2 cleavage are highly selectively blocked by NH-23-C2. However, caspase-3 and caspase-8 are free from NH-23-C2 blockage, which makes them suitable chemical tools with which to explore the exact function of caspase-2 in diverse types of biological equipment [104].

## 5. Necroptosis and Colon Cancer

### 5.1. The Mechanism of Necroptosis

As a programmed cell death, necroptosis is necrosis-like cell death mediated by caspase inhibition (Figure 4). When FasL or TNFα is stimulated, certain apoptosis-deficient types of cells will undergo necroptosis, which has features of either apoptosis or necrosis [105]. Several signaling pathways participate in necroptosis through three major proteins: receptor-interacting serine/threonine-protein kinase 1 (RIPK1), receptor-interacting serine/threonine-protein kinase 3 (RIPK3), and mixed lineage kinase ligand (MLKL) [106]. The domain dimerization of MLKL induced by RIPK3 is a critical process for MLKL activation in necroptosis [107]. Increasing evidence suggests that necroptosis is a promising target for cancer and deserves further investigation [108].

TNFR and TLR are necroptosis inducers. They lead to the recruitment of RIPK1, FADD, and caspase8/10. RIPK1 is the upstream regulator. RIPK1 is dimerized and activated after binding with FADD. Activated RIPK1 combines with and activates RIPK3. MLKL is combined and activated subsequently. The complex they form is necrosome. MLKL commits cell necroptosis. It oligomerizes and translocates to the cell membrane and causes permeabilization. It results in leakage and cell swelling and ultimately causes necroptosis.

### 5.2. Necroptosis-Related DNA and RNA of Colon Cancer

The state of p53 can make a prominent difference in the outcome of colon cancer cells in antitumor treatment. In the absence of p53, the starvation of colon cancer cells reduces ATP, and AMP-activated protein kinase (AMPK) is activated and protects cells from necroptosis. Yet, in HCT116 p53+/+ cells, AMPK cannot have the same effect [109]. Drug resistance is more likely to occur in p53−/− cells. Soyauxinium chloride (SCHL) has prominent cytotoxic potential on drug-resistant p53−/− cells. It increases ROS production and causes necroptotic cell death [68]. GSK3B can induce necroptosis in drug-resistance p53−/− colon cancer cells. Targeting GSK3B combined with chemotherapy might become a novel strategy for chemo-resistant tumors [110,111]. Apart from p53, RNA can also be a target in colon cancer therapy. It has been reported that non-coding RNA combined with Hsp90 inhibitor is a novel target therapy [112].

### 5.3. Anti-Tumor Treatments Inducing Necroptosis

As one of the key proteins in necroptosis, RIPK1 can be regulated by fragile X mental retardation protein (FMRP) [108] or silymarin (SM) [113]. They meditate polo-like kinase 1 (PLK1) [114] to control necroptotic activity in colon cancer. They can also interact with other substances such as mitochondrial Ca2+uniporter (MCU) and prompt the proliferation of colon cancer [115]. The inhibition of the N-end rule pathway, which degrades RIPK1, can promote necroptosis and lead to tumor regression [116]. RIPK1 is also known as a driver of 5-FU-induced TNF-α-dependent necroptosis [117]. RIPK1 can ensure chromosome alignment and maintain genome stability [114].

Another key protein RIPK3 plays a critical role in tumor suppression [118]. RIPK3 increases after being exposed to heat and prompts necroptosis in colon cancer cells [119]. HA-P-LP (hyaluronic acid-conjugated cationic liposomes) is a vector of mRIPK3-pDNA overexpression in tumors. HA-P-LP enhances the tumor-targeting effect in colon cancer therapy [120]. The overexpression of RIPK3 combined with chloroquine (CQ) can solve multi-drug resistance colon cancer [121]. HMA [122], the HECT domain E3 ligase HACE1 [123], the second mitochondria-derived activator of caspase (SMAC) [124], poly-C [125], CBN (columbianadin) [126], and cationic peroxidase (PmPOD) [127] induce necroptosis in colon cancer cells. Additionally, RIPK3 is the essential factor.

The third major protein MLKL, attributed to MEK/ERK activation in dendritic cells (DCs), protects colon from tumorigenesis [128]. Immunotherapy based on MLKL-mRNA can induce T cells that promptly work against tumor neo-antigens and slow down the progression of colon cancer [129]. A novel therapy relies on co-encapsulation liposomes. Co-encapsulation liposomes co-localize the molecules in need of colon cancer cells, including MLKL. This significantly raises the suppression rate of the tumor and may be a valid therapy for chemo-resistant colon cancer [130].

These proteins do not work alone. RIPK1 can phosphorylate itself and RIPK3. RIPK3 phosphorylates MLKL. They become necrosome, which comes into contact with PGAM5 and DRP1. Then, they cause membrane rupture and leakage. Finally, necroptosis is executed [131]. Necrostatin-1 induces necroptosis by preventing the interaction between RIPK1 and RIPK3 [132]. Cobalt chloride [133] and GLTP overexpression [134] rely on RIPK1, RIPK3, and MLKL to induce necroptosis in colon cancer cells. HPA3P induces necroptosis in colon cancer cells depending on RIPK3 and MLKL [135]. 2-methoxy-6-acetyl-7-methyljuglone (MAM) induces necroptosis in colon cancer cells by forming the RIPK1/RIPK3 complex, activating JNK and elevating ROS. In addition, MAM is independent of TNFα, p53, and MLKL. This feature of MAM brings novel insights into necroptosis [136].

Instead of targeting the three major proteins, PARP-1, a downstream active effector of RIPK1/RIPK3, can be activated by TRAIL and induce necroptosis in colon cancer cells. PARP-1 is a potential target for anti-tumor therapy [137]. JNK, p38, and ERK MAPKs are activated by dimethyl fumarate (DMF) and induce necroptosis in colon cancer cells [138]. Moreover, TNFR1 and FADD can be activated by polyunsaturated aldehydes (PUAs) and lead to necroptosis in colon cancer cells without harming normal cells [139].

Traditional chemotherapy seldom needs RIPK1 and RIPK3 to induce necroptosis. Instead, caspase inhibitors and/or second mitochondria-derived activators of caspase mimetic are needed to sensitize colon cancer cells to RIPK1 and RIPK3 [3]. Steroidal oximes show promise in intravenous chemotherapy [140]. Yet, more accurate pathway-targeting chemotherapies deserve further investigation, and the specific mechanisms require further study.

Although it is generally accepted that necroptotic factors are anti-tumor factors, an astonishing result was discovered recently: that RIPK1, RIPK3, and MLKL may support tumor growth. This is probably because the cells that survive the activation of necroptotic factors turn into stronger cells to deal with the damage. Additionally, cells supposed to die can secrete tumor-growth stimulators. Thus, necroptotic factors may have both positive and negative effects on colon cancer, and this needs further investment [131]. Another recent study shows that MLKL gene deletion does not make a difference in colon cancer development in mice. Thus, necroptosis may not play a role in colon cancer. On the contrary, a study indicates that RIPK3, as a colon tumor suppressor, has an anti-tumoral function [118]. This controversy also requires further studies to clarify [141].

Before MLKL transfers from the cytoplasm to the cell membrane, colon cancer cells can protect themselves by repairing the cell membrane damage from necroptosis through Flotillin-mediated endocytosis and ALIX–syntenin-1–mediated exocytosis [142].

Proteasome inhibitor Bortezomib is widely used in the treatment of multiple myeloma, yet it is not an effective treatment for HCMV-infected colon cancer because HCMV can inhibit necroptosis caused by Bortezomib through stabilizing the membrane potential of the mitochondria [143]. Inhibiting glycolysis and the production of ATP is a novel way of inducing necroptosis. 3-Bromopyruvate (3BP) works as an anti-tumor agent through this procedure [144].

Immune therapy is also associated with necroptosis. Necroptosis is a better protection than apoptosis against colon cancer in murine AH1-cells, which indicates that necroptosis is a promising ICD (immunogenic cell death) form in anti-tumor therapies [145]. WEE1 kinase inhibition can sensitize colon cancer cells to immune therapy by inhibiting the activation of the G2/M cell cycle checkpoint [146]. Interestingly, immune cells may share the susceptibility of antitumor drugs with colon cancer cells, and ASC is a candidate for it [147]. Immunogenic cell death (ICD) is represented by the release of cytokines and damage-associated molecules of dying cells. A recent study suggests necroptosis as a possible preferred ICD pattern over apoptosis in the treatment of colon cancer [145].

### 5.4. Prevention of Colon Cancer by Inhibiting Necroptosis

IBD (inflammatory bowel disease) is highly related to the tumorigenesis of colon cancer. As pro-inflammatory cell death, necroptosis contributes to the development of IBD. LY3009120 decreases the risk of colon cancer by inhibiting RAF and subsequently inhibits necroptosis and downstream inflammation [148]. RIPK3 enhances the increase of premalignant intestinal epithelial cells and relates to tumor-inducing colitis through JNK and CXCL1 signaling pathways [149]. The MTOR/RIPK3/necroptosis axis is an initiator of IBD and colon cancer. Gut epithelial TSC1/mTOR reduces necroptosis by inhibiting the expression and activation of RIPK3, thus preventing IBD and even colon cancer [150].

The truncation of the regulatory deoxyadenosine tract element (DATE) happens in all colon cancer cells that had microsatellite instability (MSI). The truncation of DATE activates the HGF gene, which reduces the level of RIPK1 through MET. It promotes necroptosis resistance and associates with pessimistic prognosis. Therefore, DATE might be a prognostic predictor for therapies that target HGF-MET signaling. Additionally, colon cancer patients who are DATE mutant carriers require closer clinical surveillance and screening to prevent colon cancer burden [151].

### 5.5. Relationship of Necroptosis with Colon Cancer Prognosis

RIPK3 expression has been found to positively relate to a better prognosis of colon cancer, which makes it a promising prognostic marker [152]. The expression of MLKL has a similar effect to RIPK3 [153]. The recurrence of colon cancer can be predicted by 26 autophagy-related genes (ARGs), in which BAX and PARP1 play the most significant role. These 26 ARGs are enriched in necroptosis pathways. Their stratification of patients can benefit the early detection of recurrence [154].

## 6. Parthanatos and Colon Cancer

### 6.1. The Mechanism of Parthanatos

As a non-canonical programmed cell death, parthanatos is featured by PARP-1 hyperstimulation [155], increasing PAR polymer [114], and the movement to the nuclear of apoptosis-inducing factor (AIF) from chondriosome. They are the primary initiators of parthanatos. Unlike pyroptosis, parthanatos does not depend on caspase. Instead, the over-reactive oxygen species (ROS) response is the initiator [156]. ROS leads to an increase in PAR polymer and the movement of AIF from chondriosome to nuclear [157]. As for the pattern of cell killing, it is observed that parthanatos can induce the dissipation of the chondriosome membrane and the vast-scale of fractured DNA chromatin condensation [158].

### 6.2. Anti-Tumor Treatments Inducing Parthanatos

ETS transcription factors are closely related to tumorigenesis. MAPK and the loss of p53 enhance the sensitivity of cancer cells against YK-4-279, the ETS factor inhibitor. YK-4-279 induces parthanatos by the hyperPARylation of PARP1, translocating AIF and leading the chondriosome membrane to depolarization [159]. Similarly, the alkylating agent N-methyl-N0-nitro-N0-nitrosoguanidine (MNNG) also induces parthanatos through activating PARP-1. Increasing intracellular calcium and ROS generation are the promoters of this procedure. Hence, specific suppressors for this pathway might be applied to anti-tumor therapy in the future [160]. In addition, parthanatos can also be a strategy for drug-resistant cancer. The sodium–hydrogen antiporter inhibitor HMA (5-N, N-hexamethylene amiloride) can lead multi-drug-resistant cells to parthanatos through a caspase-independent model meditated by acid L-DNase II [161].

## 7. Oxeiptosis and Colon Cancer

Oxeiptosis is a new type of regulated cell death. Oxeiptosis is independent of caspase. Oxeiptosis is induced by the KEAP1-PGAM5-AIFM1 pathway, which is triggered by ROS. Magnetic hyperthermia and chemotherapy show better effects in colon cancer than single therapy. The type of cell death is presumably oxeiptosis led by ROS in colon cancer, as C-PARP, caspase-3, caspase-8, and c-caspase-8 levels are steady or slightly decreased [162].

## 8. NETosis and Colon Cancer

The tumor immune microenvironment (TIME) is well known as a key role in tumor generation, progression, and treatment efficacy. Neutrophil extracellular traps (NETs) contain neutrophil DNA fibers and bind cells to promote metastatic progression. Researchers used to focus on infection-induced NETosis. Interestingly, tumors can promote neutrophils to go through NETosis without infection/surgical stress and then promote metastatic progression. Additionally, the high level of NET correlates with the cancer stage. Thus, targeting it might discourage metastatic dissemination. Thus, NET-based biomarkers help in predicting cancer progression and metastasis using a preclinical murine model of colon cancer [163].

## 9. PANoptosis Inhibits Colon Cancer

PANoptosis is activated by components of pyroptosis, apoptosis, and/or necroptosis. PANoptosis cannot be grouped into any of the three types of cell death. These three types of programmed cell death exhibit crosstalk, and they can switch from one type to another in certain circumstances. For example, the inhibition of apoptosis element CASP8 promotes RIPK3 activity leading to necroptosis, and vice versa. CASP8 and necroptosis regulator FADD both can regulate pyroptosis element NLRP3. In addition, CASP3 and CASP7 block pyroptosis by incising GSDMD. The evidence shows the connection between the three types of programmed cell death [164]. Inducing PANoptosis can be a novel way to inhibit colon cancer. It is found that TNF-α and IFN-γ alone or combined with cytokine can induce PANoptosis in human colon cancer cells. Therefore, TNF-α and IFN-γ can be therapeutically targeted [165]. PANoptosis also participates in colitis-associated tumorigenesis. As an upstream regulator, IRF (interferon regulatory factor 1) plays a role in myeloid and epithelial compartments of mice and protects them from tumorigenesis by inducing PANoptosis. Yet, more experiments on IRF in human cells are required [164].

## 10. Entosis and Colon Cancer

Some epithelial cells enter into other adjacent cells and are degraded by lysosomes or released. This cell-in-cell (CIC) interaction is entosis [166]. This phenomenon is discovered in colon cancer cells. The TRAIL (TNF-related apoptosis-inducing ligand) can initiate apoptosis by promoting the extrinsic pathway. Yet, only a portion of colon cells exposed to TRAIL will undergo apoptosis. This phenomenon means that there is a survival pathway TRAIL can induce. TRAIL can induce not only apoptosis but also entosis. Normally, the inner cells will be degraded by the outer cells. However, sometimes inner cells can go through cell division inside the outer cell or can be released. This procedure makes it understandable that CIC structures in the invasive front regions of colon tumor indicate unfavorable prognosis [167].

## 11. Summary and Future Directions

Non-canonical programmed cell death plays a significant role in colon cancer generation, progression, prevention, treatments, metastasis, and prognosis. However, there was no thorough overview of non-canonical programmed cell death and colon cancer. Thus, our review summarized the mechanisms and future application of non-canonical programmed cell death in relation to different aspects of colon cancer.

Drug resistance is often an important obstacle in tumor therapy. For example, GW4064 enhances the chemotherapy sensitivity of colorectal cancer to oxaliplatin by inducing pyroptosis and reducing the resistance to oxaliplatin. In this case, pyroptosis becomes efficient combined with oxaliplatin. It indicates that pyroptosis has the potential to enhance the effect of anti-tumor treatment and reduce drug resistance. Thus, inducing pyroptosis may be a promising solution. Yet, articles about inducing pyroptosis combined with other drugs are still not enough. In addition, the role of pyroptosis in the prognosis and metastasis of colon cancer is still uncertain [101]. Mitotic catastrophe, parthanatos, ferroptosis, pyroptosis, and necroptosis can also be applied in drug-resistant tumor therapy. Such mechanisms and their clinical application deserve further study.

Sometimes non-canonical programmed cell death promotes the development of CAC. Yet inducing non-canonical programmed cell death in cancer cells might be a valid treatment. It seems paradoxical but it is actually not. Non-canonical programmed cell death in normal cells can lead to abnormal mucosal inflammation, leading in turn to tumorigenesis. In cancer cells, non-canonical programmed cell death can inhibit tumor progression and enhance the effect of other treatments.

Different types of non-canonical programmed cell death may share a common procedure. For example, some types require ROS generation, including parthanatos, ferroptosis, necroptosis, pyroptosis, and oxeiptosis (Figure 5). Cross talk between necroptosis and pyroptosis has been discovered in a recent study [168]. The evidence of interaction can also be found in PANoptosis. Further studies about the complex interactions between them are needed to better unravel the mechanisms in colon cancer.

The yellow arrows represent the ferroptosis pathway. Fe^3+^ is transferred into the cell through TFRC (transferrin receptor protein 1) and turns into Fe^2+^. Fe^2+^ constitutes LIP (labile iron pool) and increases ROS. Then, ferroptosis is induced. The green arrows represent necroptosis pathway. RIPK1/MLKL and ROS promote each other and induce necroptosis. The orange arrows represent the parthanatos pathway. MNNG or OGD induces the generation of ROS and leads to parthanatos. The blue arrows represent the pyroptosis pathway. Nicotine enters into the cell and raises ROS. ROS activates NLRP3 inflammasome leading to the activation of caspase-1. Additionally, caspase-1 contributes to pyroptosis. The purple arrows represent oxeiptosis. Increased intracellular ROS oxidizes KEAP1, and oxeiptosis is induced.

Despite recent advances in research, studies on the relationship between non-canonical programmed cell death and colon cancer metastasis are inadequate. The prevention of metastasis through inducing non-canonical programmed cell death could be a valid therapy. Not only neutrophils but also other immune cells related to non-canonical programmed cell death can be promising paths.

In conclusion, the prospect of non-canonical programmed cell death in the prevention, treatment, and prognosis prediction of colon cancer is promising. Multiple treatments can induce or combine with non-canonical programmed cell death, the application of which requires future studies to confirm.

## Figures and Tables

**Figure 1 cancers-14-03309-f001:**
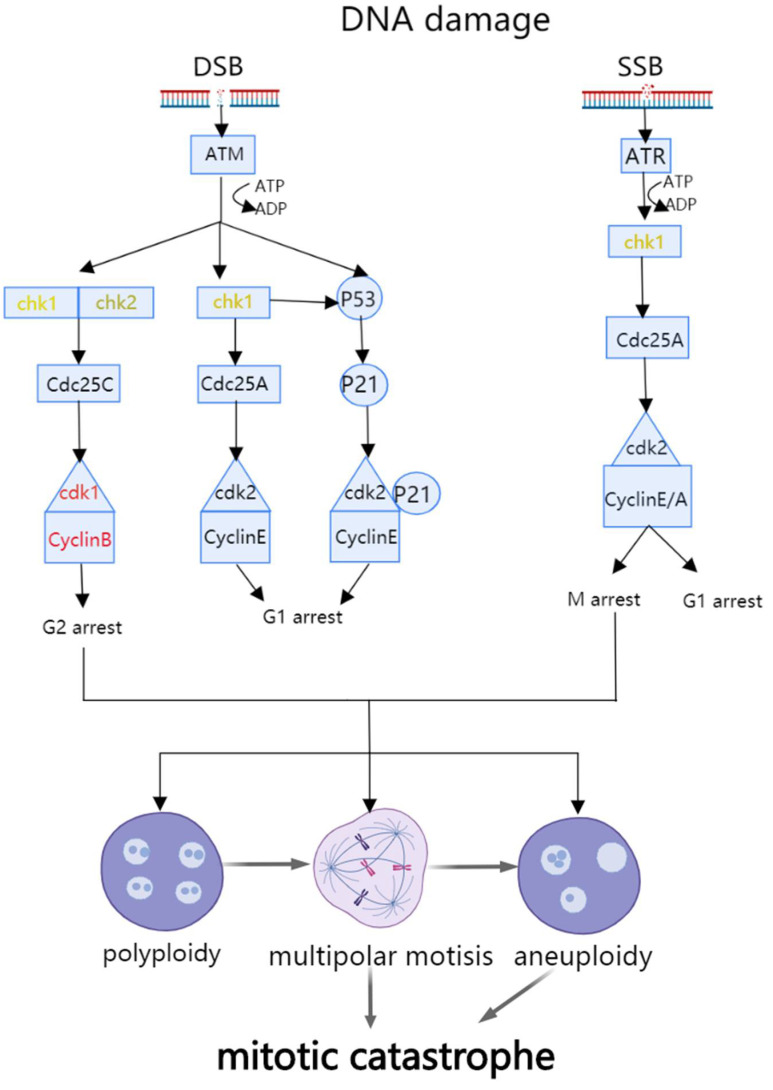
Schematic diagram of mitotic catastrophe pathway.

**Figure 2 cancers-14-03309-f002:**
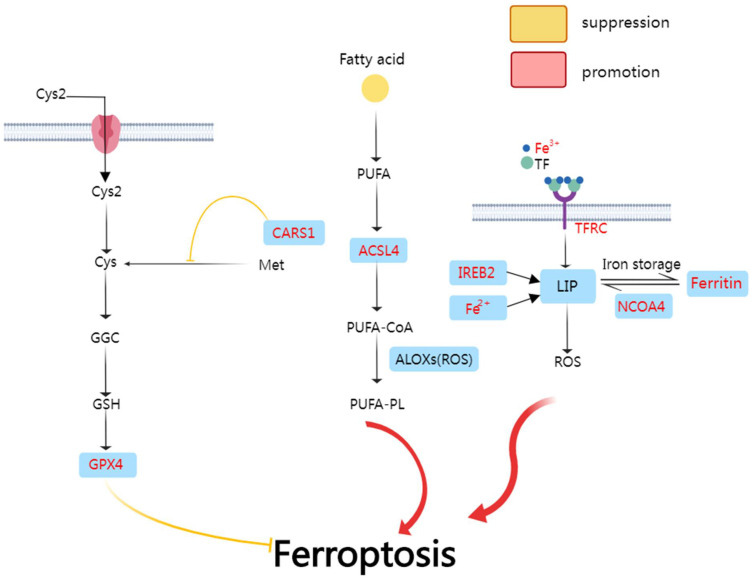
Schematic diagram of ferroptosis pathway.

**Figure 3 cancers-14-03309-f003:**
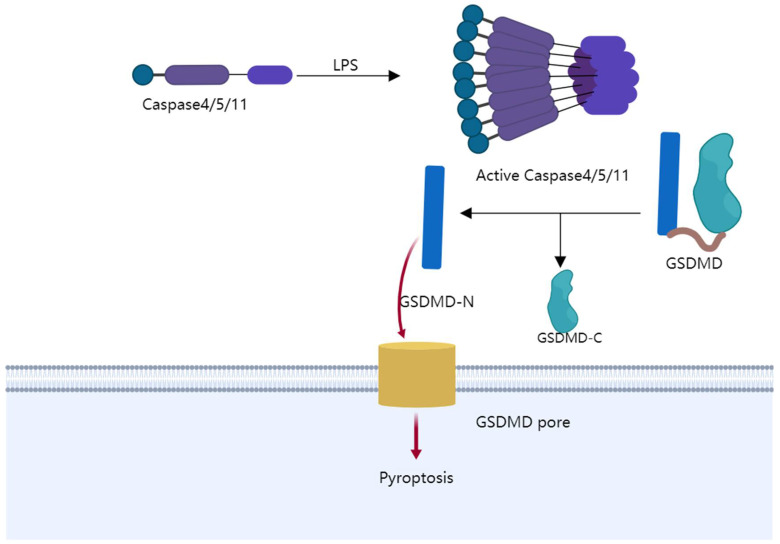
Schematic diagram of pyroptosis pathway.

**Figure 4 cancers-14-03309-f004:**
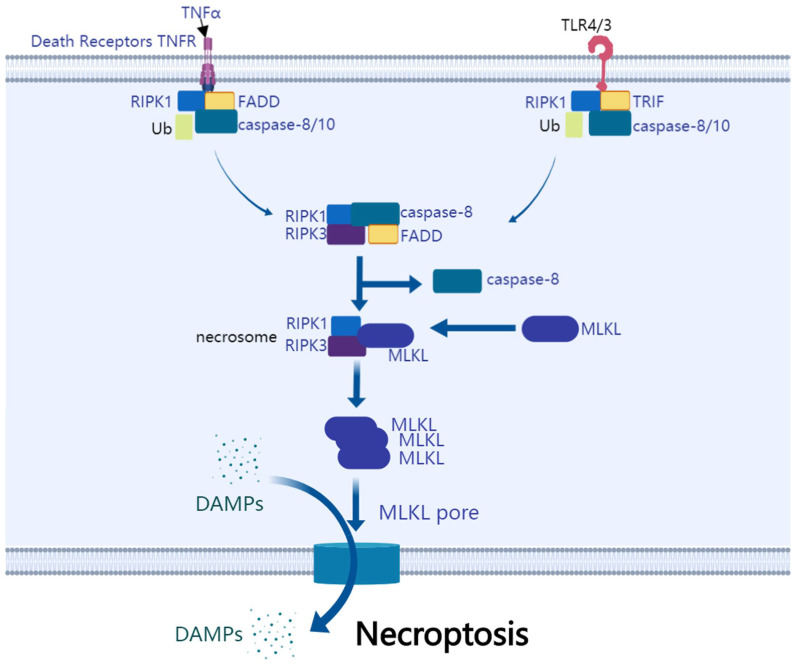
Schematic diagram of necroptosis pathway.

**Figure 5 cancers-14-03309-f005:**
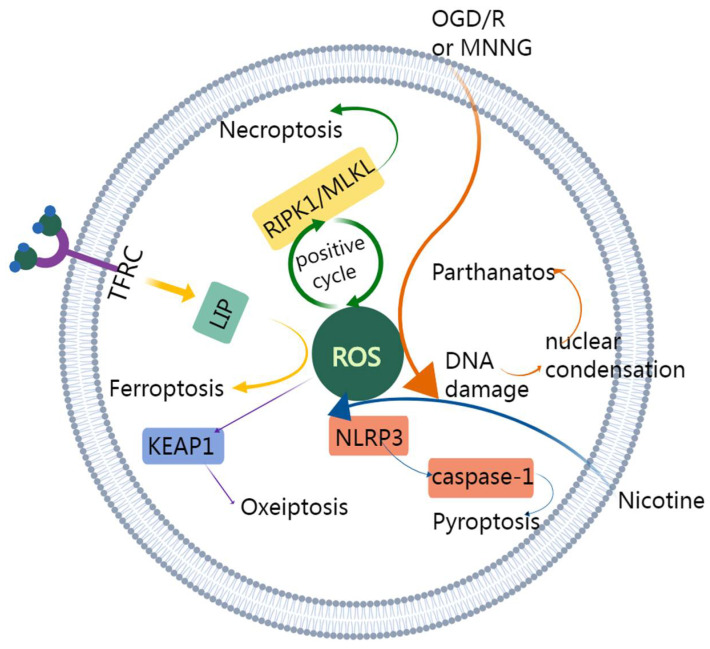
Schematic diagram of ROS in parthanatos, ferroptosis, necroptosis, pyroptosis, and oxeiptosis.

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
