# Peer review of "Non-Canonical Programmed Cell Death in Colon Cancer"

_cancers, 2022, doi:10.3390/cancers14143309_

Round 1
Reviewer 1 Report
This review article addresses the role of non-canonical cell death processes in colon cancer. Although the subject of the review is timely for an overview of the present litterature, the
the quality of the text is poor.
Major issues are:
1-Apart from the Symmary section, the text is difficult to comprehend firstly because of a high number of English grammar mistakes.
2- The structure of the presentation lacks organization, harmonizat,on nd equal distribution of the content between subtopics. Some matters are unnecessary detailed while some others are oversimplified.
3-Many abbreviations lack proper full description.
4-Well known mechanisms of different programmed cell death pathways have not been described. For example, Figure 1 entitled Schematic diagram of mitotic catastrophe pathway does not Show the events and molecular leading cells from G2 or M arrest and mitotic catastrophe. Figure 2 entitled Schematic diagram of ferroptosis pathway does not show how Iron is involved in the process of ferroptosis. Another example is Figure 4 entitled Schematic diagram of necroptosis pathway. It is unclear how RIPK and MLKL complex contributes to the necroptotic cell death.
Reviewer 2 Report
The authors did an extensive review of non-canonical programmed cell death in colon cancer. This work discusses the various molecules involved in apoptosis with carcinogenesis and the interaction with treatments. However, the article is difficult to read and not well structured, it needs a thorough review in English. Some sentences are not even understandable. In addition, the work is divided into titles and subtitles, however, some subtitles have no content and refer only to the work of a reference and do not even discuss it as an example, 2.4, 3.2.3., 5.5, 8, 9 and 10. Additionally, being immunotherapy so important in colon cancer treatment, it should be expected more explored in this article, see 4.2.3.
The authors refer to ‘colon cancer is one of the most common malignant tumors with high incidence worldwide’ but taking into account the content of this work it would be desirable to refer to numbers since colon cancer is the 3rd most incident and the 4th most mortal. Moreover, the image caption should be more descriptive, and allow the reader to understand the image with your text. The majority of the studies discussed in this work are studies conducted in colon cancer cell lines and it will be relevant for the discussion of more results in pre-clinical studies or even clinical studies. Furthermore, it will be nice to have a figure to integrate all the discussions of this work.
Reviewer 3 Report
In the review manuscript titled, “Non-canonical Programmed Cell Death In Colon Cancer” the authors provide an update on the non-canonical programmed cell death (PCD) in colon cancer. The manuscript has numerous flaws. Few examples are below.
· Authors need to expand the introduction section with citation of published literature.
· Published literature have reported that mitotic catastrophe can have a dual role in tumorigenesis. Authors should discuss this in the manuscript.
· Is CELF2 tumor suppressor gene the only gene involve in mitotic catastrophe mediated regulation of colon cancer initiation and progression. The authors expand on the subsection 2.2.
· Authors should perform comprehensive literature search and expand subsection 2.4
· Authors should also discuss the genes regulating ferroptosis, pyroptosis, and necroptosis in colon cancer in a separate subsection for each type of PCD
· The authors are recommended to reduce the number of sub-sections in each section, and each section should have more or less similar subsection headings. This will streamline the manuscript and help the reader to easily follow the manuscript content.
· The manuscript is difficult to follow in some areas. Authors are recommended to check the flow of the manuscript and to consult an expert.
Round 2
Reviewer 1 Report
Acceptable.
Reviewer 2 Report
After major reviews, the article could be accepted in its present form